

# Incidence, mortality, and predictive factors associated with acute respiratory distress syndrome in multiple trauma patients living in high-altitude areas: a retrospective study in Shigatse

Dan Tu[1,*], Lv Ji[1,*], Qiang Cao[2], Tin Ley[3], Suolangpian Duo[4], Ningbo Cheng[3], Wenjing Lin[3], Jianlei Zhang[3], Weifeng Yu[2,5], Zhiying Pan[2,3] and Xiaoqiang Wang[2,5]

[1] Department of Intensive Care Unit, Shigatse People's Hospital, Shigatse, Xizang, China
[2] Department of Anesthesiology, Renji Hospital, Shanghai Jiaotong University School of Medicine, Shanghai, Shanghai, China
[3] Department of Anesthesiology, Shigatse People's Hospital, Shigatse, Xizang, China
[4] Department of Emergency, Shigatse People's Hospital, Shigatse, Xizang, China
[5] Key Laboratory of Anesthesiology (Shanghai Jiao Tong University), Ministry of Education, Shanghai, China
* These authors contributed equally to this work.

Corresponding authors
Zhiying Pan, ssmupzy@163.com
Xiaoqiang Wang,
doctor.wxq@outlook.com

## ABSTRACT

**Background:** Acute respiratory distress syndrome (ARDS) is a severe complication that can lead to fatalities in multiple trauma patients. Nevertheless, the incidence rate and early prediction of ARDS among multiple trauma patients residing in high-altitude areas remain unknown.

**Methods:** This study included a total of 168 multiple trauma patients who received treatment at Shigatse People's Hospital Intensive Care Unit (ICU) between January 1, 2019 and December 31, 2021. The clinical characteristics of the patients and the incidence rate of ARDS were assessed. Univariable and multivariable logistic regression models were employed to identify potential risk factors for ARDS, and the predictive effects of these risk factors were analyzed.

**Results:** In the high-altitude area, the incidence of ARDS among multiple trauma patients was 37.5% (63/168), with a hospital mortality rate of 16.1% (27/168). Injury Severity Score (ISS) and thoracic injuries were identified as significant predictors for ARDS using the logistic regression model, with an area under the curve (AUC) of 0.75 and 0.75, respectively. Furthermore, a novel predictive risk score combining ISS and thoracic injuries demonstrated improved predictive ability, achieving an AUC of 0.82.

**Conclusions:** This study presents the incidence of ARDS in multiple trauma patients residing in the Tibetan region, and identifies two critical predictive factors along with a risk score for early prediction of ARDS. These findings have the potential to enhance clinicians' ability to accurately assess the risk of ARDS and proactively prevent its onset.

# INTRODUCTION

The acute respiratory distress syndrome (ARDS) was first reported in 1967, and the diagnostic criteria were created in 1992 (*Ashbaugh et al., 1967*; *Bernard et al., 1994*). Subsequently, the new Berlin Definition became widely utilized by clinical physicians for the evaluation and diagnosis of ARDS (*Ranieri et al., 2012*). As a prevalent complication, ARDS seriously affects patients' health, and leads to poor prognosis or even mortality (*Beitler et al., 2022*). A prospective analysis (*Rubenfeld et al., 2005*) of ARDS incidence reported an annual occurrence of 190,000 cases in USA, while the hospital mortality rate reached 38.5%. In the LUNG-SAFE study, researchers evaluated the incidence and outcomes of ARDS in over 29,000 patients from 50 countries (*Bellani et al., 2016*). Results showed the prevalence of ARDS reached 10% among ICU patients and 23% among those receiving ventilation. Additionally, the hospital mortality rates surprisingly reached 46.1% for those with severe ARDS. Moreover, ARDS treatments bring heavy financial burden, with data indicating mean total hospitalization costs of 12,336 USD (*Adhikari et al., 2010*; *Oh & Song, 2022*). In summary, the high morbidity and mortality of ARDS not only bring heavy burden on global health but also threaten patient prognosis.

Numerous risk factors have been identified in association with the development of ARDS, including major trauma (both surgical and accidental), pneumonia, sepsis, and others (*Matthay et al., 2019*; *Pham & Rubenfeld, 2017*). Despite significant advancements have achieved in ARDS research, further investigation is needed to establish effective preventive measures for this condition (*Meyer, Gattinoni & Calfee, 2021*; *Ranieri et al., 2012*). Previous studies have successfully developed specific scoring models for predicting the risk of postoperative complications including ARDS (*Arozullah et al., 2001*; *Gajic et al., 2011*; *Kor et al., 2014*). These models help clinical physicians quickly identify patients at high risk of developing ARDS, thereby improving patient prognosis.

According to reports, ARDS has emerged as a significant contributor to morbidity and mortality in trauma cases, with mortality rates reaching up to 40% (*Fan, Brodie & Slutsky, 2018*). In another retrospective study, researchers reported that over 51% of multiple trauma patients developed ARDS (*Haider et al., 2017*). However, treatment options for ARDS remain limited, primarily relying on supportive mechanical ventilation (*Fan, Brodie & Slutsky, 2018*; *Rizzo et al., 2023*). Consequently, early identification of patients at risk for ARDS development is crucial, and extensive research studies have focused on the identification of risk factors.

Studies have indicated that patients residing at altitudes above 1,500 m exhibit lower partial pressure of oxygen ($PaO_2$) and arterial oxygen saturation ($SaO_2$) values compared to those living at sea level (*Ortiz et al., 2022*). This can be attributed to the gradual decrease in atmospheric pressure and oxygen content in the air with increasing altitude. Consequently, individuals in high-altitude areas have undergone physiological adaptations

such as enhanced oxygen delivery, and adjustments in oxygen utilization and metabolism (*Guo et al., 2023*). These adaptations result in significant differences in respiratory physiology and pathology between patients residing in high-altitude and low-altitude regions (*Guo et al., 2023*; *Ortiz et al., 2022*). However, there is a lack of reports focusing on the incidence and mortality of ARDS, as well as potential risk factors for early prediction of ARDS in multiple trauma patients living in high-altitude areas.

The Qinghai-Tibet Plateau, known for its average elevation exceeding 4,000 m, holds the distinction of being the world's highest plateau and is home to over nine million people in China. In an effort to investigate the incidence and mortality of ARDS among multiple trauma patients residing in high-altitude regions, we conducted a retrospective study at Shigatse People's Hospital, situated at an average altitude exceeding 4,000 m. Furthermore, we meticulously screened several risk factors and identified two significant factors, enabling us to develop a predictive model for early detection of ARDS.

## METHODS

### Study design

This retrospective, single-center cohort study was conducted at Shigatse People's Hospital in Tibet, China. The study received approval from the Ethics Committee of Shigatse People's Hospital (No. 2023RKZRMYY12M005) and adhered to the Helsinki Declaration as well as the Strengthening the Reporting of Observational Studies in Epidemiology (STROBE) criteria. Written informed consent was waived by the Ethics Committee of Shigatse People's Hospital.

### Participants and grouping

Patients with multiple trauma and received treatments at Shigatse People's Hospital were enrolled between January 1, 2019, and December 31, 2021. Patients would be excluded including: (1) missing or unavailable data of patients, (2) rejection of signing informed consent and (3) patients were died before treatments. These patients were categorized into either the ARDS group or the No-ARDS group based on the diagnosis of ARDS. The diagnosis of ARDS was made by experienced ICU physicians in accordance with the Berlin Definition and the consensus conference on diagnostic criteria of ARDS at high altitudes in Western China (*Ranieri et al., 2012*; *Zhang et al., 2001*).

### Variables and outcomes

Demographic characteristics, treatment procedures, and injury severity information of patients were collected upon admission and during the course of treatment. The severity of injury was assessed using the Injury Severity Score (ISS) (*Baker et al., 1974*) and the simplified Acute Physiology Score and Chronic Health Evaluation Score (APACHE II) (*Knaus et al., 1985*). Data collection was performed by two trained researchers who extracted the necessary information from the digital medical system or paper medical records. Data were carefully checked for accuracy and entered into either Excel or the EpiData system.

### Establishment and validation of the prediction model

A logistic regression model was utilized to screen potential risk factors of ARDS. Factors with $P$-values < 0.05 in the univariable logistic regression were selected for the multivariable logistic regression, and factors with $P$-values < 0.05 in the multivariable logistic regression were selected for model construction. The risk score was calculated using the following formula: risk score = $(factor\ 1\ \times\ a) + (factor\ 2 \times b)\ldots$ + $(factor\ 3 \times n)$, where letters "a", "b", and "n" represent the regression coefficients.

### Statistical analysis

The statistical analyses were conducted using IBM SPSS Statistics software, version 23.0 (IBM SPSS Inc., Armonk, NY, USA), and the graphical representations were designed accordingly. Categorical variables were presented as numbers and percentages, while continuous variables were reported as mean and standard deviation or median (25% interquartile range, 75% interquartile range) depending on their distribution. Continuous variables were compared using either the Student's t-test or the Mann-Whitney U test, while categorical variables were compared using the Chi-squared $\chi^2$ test.

To evaluate the predictive value, a receiver operating characteristic (ROC) curve was constructed, and the area under the curve (AUC) was calculated. All statistical tests were two-sided, and $P$-values less than 0.05 were considered statistically significant.

## RESULTS

In the retrospective study, 186 patients were screened and 18 patients were excluded. Finally, a total of 168 multiple trauma patients were ultimately included (Fig. 1), and their clinical characteristics are presented in Table 1. The majority of these patients sustained injuries as a result of traffic accidents or altercations. Among multiple trauma patients residing in high-altitude areas, the incidence of ARDS was found to be 39.3% (66 out of 168), while the hospital mortality rate was 16.1% (27 out of 168). Additionally, 65.5% (110 out of 168) of patients were diagnosed with craniocerebral trauma, and 53.6% (90 out of 168) were diagnosed with thoracic injuries.

### Risk factors analysis of ARDS

The patients included in the study were divided into two groups: the ARDS group ($n = 66$) and the No-ARDS group ($n = 102$) based on the diagnosis of ARDS. A comparison of characteristics between these two groups is presented in Table 2. The findings indicate that patients in the ARDS group had significantly higher ISS scores ($P < 0.001$), a higher percentage of thoracic injuries ($P < 0.001$), a longer time from injury to hospital admission ($P = 0.007$), a higher level of AST level ($P = 0.039$), a longer duration of mechanical ventilation in the ICU ($P = 0.030$), and a higher hospital mortality rate ($P = 0.006$) compared to the No-ARDS group. Compared to the No-ARDS group, patients in the ARDS group need longer time of mechanical ventilation (the median length was 2 days in the ARDS group $vs$ 1 day in the No-ARDS group), which suggested that respiratory support was important for patients with ARDS. Specifically, the hospital mortality rate was 25.8% (17 out of 66) in the ARDS group, while it was 9.8% (10 out of 102) in the No-ARDS

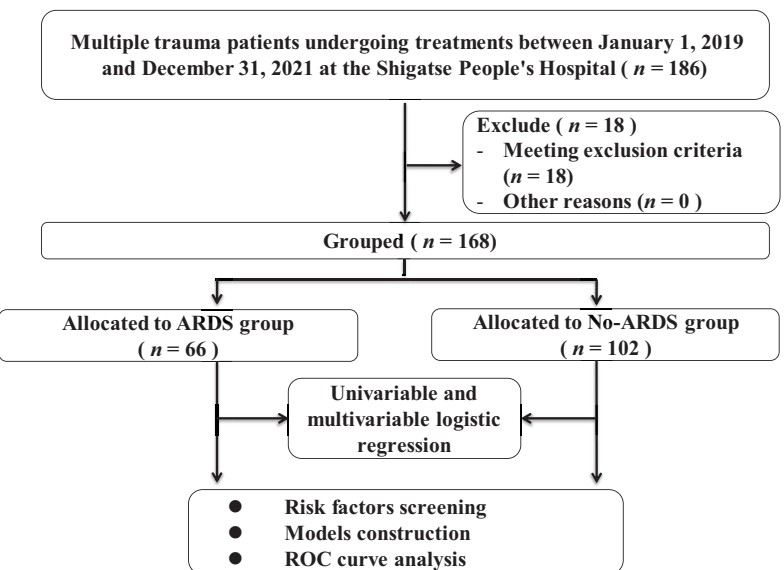

**Figure 1** The flowchart of patients' enrollment and grouping.

**Table 1** Clinical characteristics of enrolled patients.

| Characteristics | $n = 168$ |
|---|---|
| Gender (male/female) | 153/15 (91.1/8.9) |
| Age (year) | 43.47 (16.29) |
| Height (cm) | 164.60 (19.28) |
| Smoking (Yes/No) | 19/149 (11.3/88.7) |
| ASA stage (I/II/III) | 143/21/4 (85.1/12.5/2.4) |
| Chronic disease* (Yes/No) | 26/142 (15.5/84.5) |
| Pulmonary complications before surgery** (Yes/No) | 12/156 (7.1/92.9) |
| ISS score | 19.15 (8.14) |
| APACHE II score | 18.91 (7.80) |
| Trauma diagnosis*** | |
| Craniocerebral trauma (Yes/No) | 110/58 (65.5/34.5) |
| Limb trauma (Yes/No) | 43/125 (25.6/74.4) |
| Thoracic injuries (Yes/No) | 90/78 (53.6/46.4) |
| Abdominal trauma (Yes/No) | 53/115 (31.5/68.5) |
| Time from injury to hospital (hour) | 5.00 (3.00, 9.00) |
| HB after hospital admission (g/L) | 127.23 (36.43) |
| White blood cells ($10^9$/L) | 13.19 (9.72, 16.87) |
| Red blood cells ($10^{12}$/L) | 4.24 (3.41, 5.03) |
| Platelet ($10^9$/L) | 146.89 (70.54) |
| ALT (U/L) | 41.00 (23.00, 93.50) |
| AST (U/L) | 62.00 (37.00, 115.50) |
| TBIL (μmol/L) | 14.60 (8.85, 21.00) |

(Continued)

| Characteristics | $n = 168$ |
|---|---|
| Blood loss (mL) | 500.00 (400.00, 1,000.00) |
| Blood transfusion (mL) | 400.00 (300.00, 550.00) |
| Liquid transfusion (mL) | 2,100.00 (1,500.00, 2,500.00) |
| Urine (mL) | 2,115.57 (1,295.82) |
| Mechanical ventilation in ICU (day) | 1.00 (1.00, 3.00) |
| Length in ICU (day) | 6.00 (3.00, 12.00) |
| ARDS (Yes/No) | 66/102 (39.3/60.7) |
| Mild | 37 (56.1) |
| Moderate | 17 (25.8) |
| Severe | 12 (18.1) |
| In-hospital death (Yes/No) | 27/141 (16.1/83.9) |

**Notes:**
Variables are shown as "mean (SD)", "number (%)" or "median (25% quartile, 75% quartile)". ASA, American Society of Anesthesiologists; ISS, injury severity score; HB, hemoglobin; ALT, alanine transaminase; AST, aspartate aminotransferase; TBIL, total bilirubin; ICU, intensive care unit; ARDS, Acute Respiratory Distress Syndrome.
\* Chronic disease includes coronary heart disease, arrhythmia, hypertension, diabetes and hyperlipidemia.
\*\* Pulmonary complications before surgery includes phthisis, pneumonia and chronic obstructive pulmonary disease.
\*\*\* Patients with multiple traumas were counted separately.

**Table 2 Comparisons of basic characteristics between ARDS group and no-ARDS group.**

| Characteristics | ARDS group ($n = 66$) | No-ARDS group ($n = 102$) | $P$ value |
|---|---|---|---|
| Gender (male/female) | 61/5 | 92/10 | 0.621 |
| ASA grade | | | |
| I | 55 | 88 | 0.334 |
| II | 8 | 13 | |
| III | 3 | 1 | |
| Age (year) | 44.15 (16.15) | 43.04 (16.44) | 0.668 |
| Height (cm) | 166.43 (15.94) | 163.43 (21.13) | 0.328 |
| Smoking (Yes/No) | 6/60 | 13/89 | 0.465 |
| Chronic disease (Yes/No) | 12/54 | 14/88 | 0.435 |
| Pulmonary complications (Yes/No) | 8/58 | 4/98 | 0.064 |
| ISS score | **23.35 (7.93)** | **16.43 (7.07)** | **<0.001** |
| APACHE II score | 18.45 (7.31) | 19.21 (8.13) | 0.543 |
| Trauma diagnosis | | | |
| Craniocerebral trauma (Yes/No) | 42/24 | 68/34 | 0.687 |
| Limb trauma (Yes/No) | 20/46 | 23/79 | 0.261 |
| Thoracic injuries (Yes/No) | **55/11** | **35/67** | **<0.001** |
| Abdominal trauma (Yes/No) | 18/48 | 35/67 | 0.337 |
| Time from injury to hospital (hour) | **8.00 (5.00, 12.00)** | **5.50 (3.00, 9.00)** | **0.007** |
| HB after hospital admission (g/L) | 130.00 (32.23) | 125.40 (39.00) | 0.508 |
| White blood cells ($10^9$/L) | 12.13 (9.29, 16.60) | 13.79 (10.10, 17.64) | 0.126 |

| Characteristics | ARDS group (n = 66) | No-ARDS group (n = 102) | P value |
|---|---|---|---|
| Red blood cells ($10^{12}$/L) | 4.17 (3.52, 4.90) | 4.35 (3.33, 5.06) | 0.850 |
| Platelet ($10^9$/L) | 152.24 (68.91) | 143.36 (71.72) | 0.429 |
| ALT (U/L) | 51.50 (27.25, 92.75) | 38.00 (21.00, 103.00) | 0.107 |
| AST (U/L) | **77.00 (46.50, 120.00)** | **55.00 (34.00, 111.00)** | **0.039** |
| TBIL (μmol/L) | 14.90 (9.88, 21.00) | 14.20 (8.60, 21.40) | 0.748 |
| Blood loss (mL) | 500.00 (300.00, 1,300.00) | 500.00 (400.00, 800.00) | 0.884 |
| Blood transfusion (mL) | 300.00 (300.00, 500.00) | 400.00 (300.00, 700.00) | 0.057 |
| Liquid transfusion (mL) | 2,200.00 (1,700.00, 2,500.00) | 2,050.00 (1,500.00, 2,575.00) | 0.520 |
| Urine (mL) | 2,131.93 (1,084.01) | 2,107.08 (1,399.63) | 0.921 |
| Mechanical ventilation in ICU (day) | **2.00 (1.00, 5.00)** | **1.00 (1.00, 2.00)** | **0.030** |
| Length in ICU (day) | 6.00 (3.00, 13.25) | 6.00 (2.00, 12.00) | 0.774 |
| In-hospital death (Yes/No) | **17/49** | **10/92** | **0.006** |
| Mild ARDS | **4/33** | | |
| Moderate ARDS | **7/10** | | |
| Severe ARDS | **6/6** | | |

**Note:**
Variables are shown as "mean (SD)" or "median (25% quartile, 75% quartile)". ASA, American society of anesthesiologists; ISS, injury severity score; HB, hemoglobin; ALT, alanine transaminase; AST, aspartate aminotransferase; TBIL, total bilirubin; ICU, intensive care unit; ARDS, Acute Respiratory Distress Syndrome. Factors with $P < 0.05$ are labeled as bold font.

**Table 3 Univariable and multivariable logistic analyses of risk factors for ARDS.**

| Characteristics | Univariable logistic analysis | | | Multivariable logistic analysis | | |
|---|---|---|---|---|---|---|
| | OR | 95% CI | P value | OR | 95% CI | P value |
| Gender (male/female) | 0.75 | [0.25–2.31] | 0.622 | | | |
| ASA grade | 1.40 | [0.70–2.81] | 0.349 | | | |
| Smoking (Yes/No) | 1.46 | [0.53–4.06] | 0.467 | | | |
| Chronic disease (Yes/No) | 1.40 | [0.60–3.24] | 0.437 | | | |
| Pulmonary complications (Yes/No) | 3.38 | [0.98–11.72] | 0.055 | | | |
| Age (year) | 1.00 | [0.99–1.02] | 0.666 | | | |
| Height (cm) | 1.01 | [0.99–1.03] | 0.335 | | | |
| ISS score | **1.13** | **[1.08–1.18]** | **<0.001** | **1.10** | **[1.05–1.16]** | **<0.001** |
| APACHE II score | 0.99 | [0.95–1.03] | 0.541 | | | |
| Trauma diagnosis | | | | | | |
| Craniocerebral trauma (Yes/No) | 0.88 | [0.46–1.67] | 0.687 | | | |
| Limb trauma (Yes/No) | 1.49 | [0.74–3.01] | 0.262 | | | |
| Thoracic injuries (Yes/No) | **9.57** | **[4.45–20.58]** | **<0.001** | **6.84** | **[3.07–15.23]** | **<0.001** |
| Abdominal trauma (Yes/No) | 0.72 | [0.36–1.42] | 0.338 | | | |
| Time from injury to hospital (hour) | 1.03 | [1.00–1.06] | 0.102 | | | |
| HB after hospital admission (g/L) | 1.00 | [1.00–1.01] | 0.429 | | | |
| White blood cells ($10^9$/L) | 0.97 | [0.93–1.01] | 0.186 | | | |

(Continued)

| Characteristics | Univariable logistic analysis | | | Multivariable logistic analysis | | |
| --- | --- | --- | --- | --- | --- | --- |
| | OR | 95% CI | P value | OR | 95% CI | P value |
| Red blood cells ($10^{12}$/L) | 1.00 | [0.95–1.04] | 0.855 | | | |
| Platelet ($10^9$/L) | 1.00 | [1.00–1.01] | 0.427 | | | |
| ALT (U/L) | 1.00 | [1.00–1.00] | 0.637 | | | |
| AST (U/L) | 1.00 | [1.00–1.00] | 0.211 | | | |
| TBIL (µmol/L) | 1.00 | [0.97–1.03] | 0.954 | | | |
| Blood loss (mL) | 1.00 | [1.00–1.00] | 0.239 | | | |
| Blood transfusion (mL) | 1.00 | [1.00–1.00] | 0.066 | | | |
| Liquid transfusion (mL) | 1.00 | [1.00–1.00] | 0.422 | | | |
| Urine (mL) | 1.00 | [1.00–1.00] | 0.920 | | | |
| Mechanical ventilation in ICU (day) | 1.05 | [0.97–1.13] | 0.245 | | | |
| Length in ICU (day) | 1.02 | [0.98–1.05] | 0.387 | | | |

**Note:**
ASA, American Society of Anesthesiologists; ISS, injury severity score; HB, hemoglobin; ICU, intensive care unit; ARDS, Acute Respiratory Distress Syndrome. Factors with $P < 0.05$ are labeled as bold font.

group, indicating a significant increase in mortality in the ARDS group. In addition, it showed that patients with mild ARDS had lower mortality (10.8%) compared to moderate ARDS (41.2%) and severe ARDS (50%), which revealed that higher ARDS grading may lead to worse prognosis for patients.

Subsequently, a logistic regression model was employed to identify potential risk factors for ARDS. Two variables, ISS score and thoracic injuries, demonstrated a significant association with ARDS in the univariable logistic regression analysis ($P < 0.05$). These two factors were then integrated into a multivariable logistic regression model (Table 3), which further confirmed their significant correlation with the incidence of ARDS. Based on the results of the multivariable logistic regression analysis, a risk score was calculated using the following formula:

$$\text{Risk score} = (0.096 \times \textit{ISS score}) + (1.922 \times \textit{chest trauma})$$

## ROC curves and AUC analysis

To evaluate the predictive capacity of the selected risk factors and the risk score model, ROC curves and AUC analysis were conducted. As depicted in Fig. 2, both the ISS score (AUC = 0.75, 95% CI [0.676–0.824]) and thoracic injuries (AUC = 0.75, 95% CI [0.669–0.821]) exhibited promising predictive value for ARDS.

Furthermore, in an effort to enhance the predictive accuracy for ARDS, we utilized the risk score model, which amalgamated the ISS score and thoracic injuries, to construct the ROC curve. As illustrated in Fig. 3, the AUC of the risk score reached 0.82 (95% CI [0.757–0.881]), indicating a substantial effectiveness in early prediction of the incidence of ARDS.

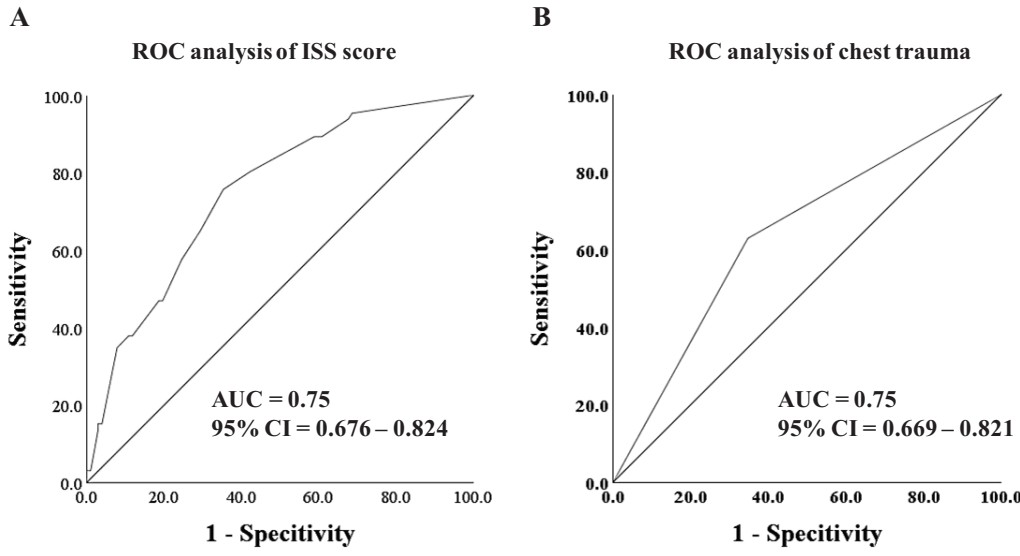

**Figure 2 ROC analysis and curves drawing.** (A) The ROC analysis of ISS score. (B) The ROC analysis of thoracic injuries.

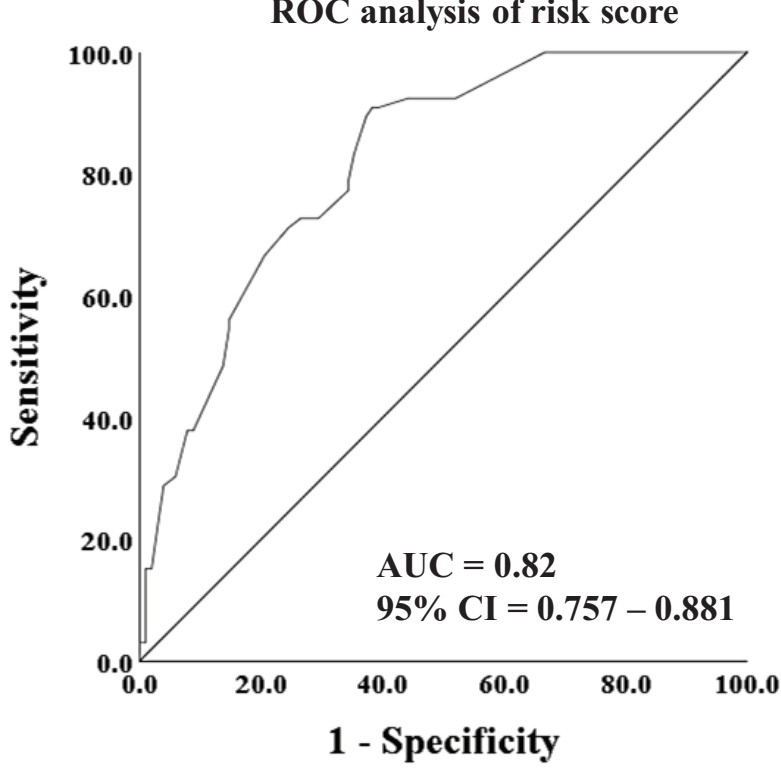

**Figure 3 ROC analysis and curves drawing of risk score.**

## DISCUSSION

ARDS is a highly detrimental complication that significantly impacts the prognosis of patients. Furthermore, the challenging plateau environment further complicates the management of ARDS in high-altitude areas, making early prediction and treatment of utmost importance for improving patient survival in clinical practice. However, limited reports exist on ARDS in multiple trauma patients specifically in high-altitude areas. In this retrospective study, we aimed to investigate the incidence and mortality of ARDS in patients with multiple trauma in the Shigatse area, characterized by an average altitude exceeding 4,000 m. Additionally, we identified several risk factors and established a risk score model for the early prediction of ARDS in these patients. The risk score model demonstrated robust predictive capabilities, with an AUC exceeding 0.8. These findings provide valuable insights for clinical physicians working in high-altitude areas, serving as a reference for their practice.

*Rodriguez Lima et al. (2023)* developed a predictive model to estimate in-hospital mortality for patients with ARDS due to COVID-19 in high-altitude regions. Their findings identified age, weight, creatinine levels, and the presence of ARDS as risk factors associated with increased patient mortality. In another study, *Liu et al. (2022)* retrospectively analyzed clinical data from patients with ARDS residing in the Qinghai area (altitude: 2,261 m). They observed a significant increase in ICU mortality corresponding to the severity of ARDS: mild (17.76%), moderate (21.43%), and severe (47.37%). According to Liu's report, the overall mortality among patients with ARDS who required mechanical ventilation reached 24.0% (55/229).

In our retrospective study, we reported the incidence of ARDS reached 39.3% in multiple trauma patients, and the hospital mortality reached 16.1%. These rates were significantly higher than those reported in studies conducted in low-altitude areas (*Kor et al., 2014*; *Milot et al., 2001*; *Sanfilippo et al., 2022*). Notably, relevant studies *Milot et al. (2001)*, *Sanfilippo et al. (2022)* investigating ARDS following cardiac surgery reported incidence rates ranging from 0.4% to 8.1%. Additionally, a previous study (*Wang et al., 2022*) conducted by our team found an ARDS incidence rate of 7.2% (74/1,032) in patients undergoing hepatectomy. In conclusion, high-altitude environments may exert significant effects on the development of ARDS.

Several studies have attempted to identify effective predictors or develop prediction models for early detection of ARDS in patients with specific diseases or injuries (*Afshar et al., 2019*; *Ding et al., 2022*; *Huang et al., 2021*). For instance, *Ding et al. (2022)* conducted a study on patients with acute pancreatitis and developed a logistic model to predict the in-hospital incidence of ARDS. They found that several independent factors, including white blood cell count, prothrombin time, ALB, serum creatinine, and triglycerides, were associated with the occurrence of ARDS during hospitalization. In a retrospective study by *O'Leary et al. (2016)* involving 305 trauma patients, clinical predictors of early ARDS were investigated. The study reported an incidence rate of 19.3% (59/305) for ARDS among trauma patients. Additionally, it highlighted that patients with blunt trauma, those who received fresh frozen plasma, or underwent thoracotomy, were at a higher risk of

developing early ARDS. These studies contribute to our understanding of potential risk factors and prediction models for the early identification of ARDS in specific patient populations.

In our study, we identified the ISS score and thoracic injuries as independent risk factors for the occurrence of ARDS. Both factors exhibited a good predictive value for ARDS, with an area under the curve (AUC) of 0.75. Furthermore, when combining the ISS score and thoracic injuries into a new risk score, we observed improved effectiveness in predicting ARDS occurrence, with an AUC of 0.82. These findings are consistent with a retrospective analysis conducted by *Haider et al. (2017)*, which also reported a significant association between the ISS score, thoracic injuries, and a higher incidence of ARDS. Specifically, thoracic trauma was identified as an important risk factor for the development of ARDS and its earlier onset in polytraumatized patients.

In summary, our results suggest that both the ISS score and thoracic injuries serve as effective indicators for predicting the occurrence of ARDS in high-altitude areas. Patients with multiple traumas, especially those with higher ISS scores or thoracic injuries, require close monitoring. Prompt measures to prevent ARDS should be implemented, such as oxygen therapy and mechanical ventilation, while fluid therapy should be judiciously administered. However, it is important to acknowledge certain limitations of our study. Firstly, due to its retrospective design, potential biases and confounding factors cannot be completely ruled out. Also, exact causality relationships between ARDS and risk factors cannot be established and further prospective studies are needed. Secondly, our study was conducted at a single center and the prediction model was only validated using internal data. Therefore, further studies are warranted to validate these results across multiple centers. Additionally, prospective studies with larger sample sizes would be necessary in the future to strengthen the evidence base.

## CONCLUSION

In conclusion, we reported the incidence of ARDS in multiple trauma patients lived in Tibetan area, and identified two critical predictive factors (ISS score and thoracic injuries) for early predicting ARDS occurrence. In addition, we constructed a risk score model for early predicting ARDS using the logistic regression, and the AUC reached 0.82. These tools may improve clinicians' ability to early estimate the risk of ARDS and timely prevent its emergence in high-altitude areas.

## LIST OF ABBREVIATIONS

| | |
|---|---|
| **ASA** | American Society of Anesthesiologists |
| **ALI** | acute lung injury |
| **ARDS** | Acute Respiratory Distress Syndrome |
| **ISS** | injury severity score |
| **APACHE** | Acute Physiology Score and Chronic Health Evaluation Score |
| **HB** | hemoglobin |
| **ALT** | alanine transaminase |
| **AST** | aspartate aminotransferase |

| | |
|---|---|
| **TBIL** | total bilirubin |
| **ICU** | intensive care unit |
| **PaO$_2$** | partial pressure of oxygen |
| **SaO$_2$** | arterial oxygen saturation |
| **SD** | standard deviation |
| **ROC** | receiver operator characteristic |
| **AUC** | area of under the curve |

### Funding

This work was supported by the Shanghai Engineering Research Center of Peri-Operative Organ Support and Function Preservation (20DZ2254200), the Shanghai 2021 "Science and Technology Innovation Action Plan" domestic science and technology cooperation project, Grant/Award Number: 21015801500, and the Postdoctoral Fellowship Program of CPSF under Grant Number 380389. The funders had no role in study design, data collection and analysis, decision to publish, or preparation of the manuscript.

### Grant Disclosures

The following grant information was disclosed by the authors:
Shanghai Engineering Research Center of Peri-Operative Organ: 20DZ2254200.
Science and Technology Innovation Action Plan: 21015801500.
CPSF: 380389.

### Competing Interests

The authors declare that they have no competing interests.

### Author Contributions

- Dan Tu performed the experiments, prepared figures and/or tables, and approved the final draft.
- Lv Ji performed the experiments, prepared figures and/or tables, and approved the final draft.
- Qiang Cao performed the experiments, prepared figures and/or tables, and approved the final draft.
- Tin Ley analyzed the data, prepared figures and/or tables, and approved the final draft.
- Suolangpian Duo analyzed the data, prepared figures and/or tables, and approved the final draft.
- Ningbo Cheng analyzed the data, prepared figures and/or tables, and approved the final draft.
- Wenjing Lin analyzed the data, prepared figures and/or tables, and approved the final draft.
- Jianlei Zhang analyzed the data, prepared figures and/or tables, and approved the final draft.

- Weifeng Yu conceived and designed the experiments, prepared figures and/or tables, authored or reviewed drafts of the article, and approved the final draft.
- Zhiying Pan conceived and designed the experiments, authored or reviewed drafts of the article, and approved the final draft.
- Xiaoqiang Wang conceived and designed the experiments, prepared figures and/or tables, authored or reviewed drafts of the article, and approved the final draft.

### Human Ethics

The following information was supplied relating to ethical approvals (*i.e.*, approving body and any reference numbers):

The Ethics Committee of the Shigatse People's Hospital.

### Data Availability

The raw measurements are available in the Supplemental File.

### Supplemental Information

Supplemental information for this article can be found online at http://dx.doi.org/10.7717/peerj.17521#supplemental-information.

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
