# Peer review of "Incidence, mortality, and predictive factors associated with acute respiratory distress syndrome in multiple trauma patients living in high-altitude areas: a retrospective study in Shigatse"

_PeerJ, doi:10.7717/peerj.17521_

## Round 0.1 · original submission · Major Revisions

Dear authors

Two reviewers have given their comments about your manuscript.

Please read carefully and respond to each concern.

Reviewer 1 ·

Basic reporting

Congratulations to the authors for the effort in developing this study.

Experimental design

It is important to recognize that the nature of this retrospective study means that a non-probabilistic sample is obtained, therefore causality relationships cannot be established, although they can be correlated and, in particular, propose hypotheses from prospective studies that allow causality to be affirmed. Authors should review these statements in the development of the article.

Validity of the findings

It is essential to explain in text and graphically the populations that developed mild-moderate-severe ARDS and the outcomes in each of these groups.

As we have a limited population (168), excluding 18 patients (10%) may have an impact on the final results. I recommend that the authors express their opinions in the text about this observation.

The model includes the days on mechanical ventilation because they have a significant statistical difference, but when reviewing the table the data are 1 versus 2 days, that is, recognizing whether the statistical difference is as important as a clinical difference. I recommend doing this with a cut-off point determined in hours; if this is not possible, the authors should explain in the text why they consider it relevant.

Additional comments

It is important to expand on the reason why the authors consider that in the population residing at high altitudes, the outcomes including the onset of ARDS and death related to it may be different.

·

Basic reporting

The writing is clear and the manuscript is well structured.

Experimental design

The study design is reasonable but could be improved. Please see the additional comments.

Validity of the findings

the conclusion is rigorous with limitations explained.

Additional comments

Tu et al. explored the incidence rate and mortality of ARDS for multi-trauma patients in high-altitude areas, analysed the associated factors of ARDS and built a risk model of ARDS with trauma patients, maintaining acceptable performance. The study design, analysis methods and the writing are clear to me. This study is valuable as more attention should be paid on the patients/people in the high-altitude. Several issues should be solved at this stage for the manuscript.

Do multiple trauma patients with multi-site traumas have a higher risk of ARDS and mortality? For example, patients with three traumas are riskier than patients with two traumas? Though it is related with ISS, an analysis is recommended.

Why other clinical markers are not included in your analysis? The author includes only HB. As the author reviewed, blood cell counts, ALB, TG and other markers (for example, blood pressure, pulse rate) are associated with ARDS as well. Markers related with lung function should be considered. Does the race of patients, Tibetan and Han Chinese, have an impact on ARDS as they are correlate with multi clinical indexes? It is worth including other accessible markers in the prediction model, which probably improve the performance of the model.

Why not include face and external traumas in the basic characteristics of the trauma patients as they are also used in the commutation of ISS?

If the author could discuss actionable operations for these trauma patients with a high risk of ARDS is favorable.

---

## Round 0.2 · accepted · Accept

Since the authors have addressed all of the reviewers' comments, I recommend accepting this manuscript version.

Reviewer 1 ·

Basic reporting

The modifications respond to the observations. I don´t have any additional comments.

Experimental design

The modifications respond to the observations. I don´t have any additional comments.

Validity of the findings

The modifications respond to the observations. I don´t have any additional comments.

Additional comments

The modifications respond to the observations. I don´t have any additional comments.

·

Basic reporting

no comment

Experimental design

no comment

Validity of the findings

no comment

Additional comments

The authors have solved my concern. I am satisfied with this new revision.